# Learning to Select Camera Views: Efficient Multiview Understanding at Few Glances

## Abstract

Multiview camera setups have proven useful in many computer vision applications for reducing ambiguities, mitigating occlusions, and increasing field-of-view coverage. However, the high computational cost associated with multiple views creates a significant challenge for end devices with limited computational resources. To address this issue, we propose a view selection approach that analyzes the target object or scenario from given views and selects the next-best-view for recognition or detection. Our approach features a reinforcement learning based camera selection module, MVSelect[1], that not only selects views but also facilitates joint training with the task network. Experimental results on multiview classification and detection tasks show that our approach achieves promising performance while using only 2 or 3 out of $N$ available views, significantly reducing computational costs. Furthermore, analysis on the selected views reveals that certain cameras can be shut off with minimal performance impact, shedding light on future camera layout optimization for multiview systems.

## 1 Introduction

Multiple camera views (multiview) are popular in computer vision systems for their ability to address challenges such as occlusions, ambiguities, and limited field-of-view (FoV) coverage. Tasks like classification (Su et al., 2015; Qi et al., 2016) and detection (Chavdarova et al., 2018; Hou et al., 2020) have shown significant benefits from using multiple cameras (Fig. 1). With reduced hardware cost and easy deployment, real-world products now include more cameras at larger scales.

However, the use of multiple cameras comes at a high computational cost, which can be a significant challenge for end devices with limited computational resources, especially with higher image resolutions and deeper neural network backbones. Limiting image resolution or using lighter networks (Molchanov et al., 2016; Howard et al., 2017) are current options to reduce computation, but they may impede the progress in camera sensors or neural network architecture.

To address this challenge, this paper introduces a new angle to efficient multiview understanding by selecting only the most useful views. To identify the best views, this approach leverages camera layouts, which is a key aspect overlooked by existing alternatives. With known camera layouts, networks should be able to infer what each camera view looks like and then choose accordingly, as previous works (Kanezaki et al., 2018) have shown that networks can associate images with camera poses. Existing work on active vision (Aloimonos et al., 1988; Findlay & Gilchrist, 2003; Chen et al., 2011) indicates that it is possible to find next-best-views for reconstruction with 3D sensors. In this paper, we focus on recognition and detection problems using multiple RGB cameras, where a side view may confuse a guitar with a mandolin (Fig. 2). However, prior knowledge of the camera layout should inform the system that the front view can clear the ambiguities and should be queried.

To achieve this goal, this paper proposes a novel view selection module, MVSelect, that chooses the best camera views from any initial view. The proposed module first analyzes the target object or scenario using the given view and then selects the next view that best helps the task (classification or detection) network. To navigate through the non-differentiable view selection (the system only looks at selected cameras, so not-selected views cannot back propagate gradients to the controller),

---

[1]Code available at `https://anonymous.4open.science/r/MVSelect-38B8`.

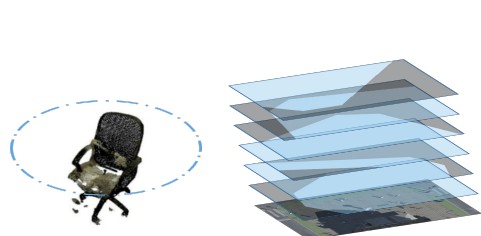

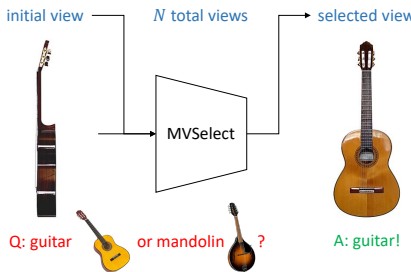

Figure 1: Example of multiview camera setups. **Left**: multiview classification jointly considers multiple camera views (blue dots) to identify the object. **Right**: multiview detection estimates pedestrian occupancy from multiple cameras (blue FoV maps) over bird's-eye-view (bottom colored image). For both classification and detection tasks, due to hardware constraints, camera layouts are usually pre-defined.

Figure 2: Efficient multiview understanding with two glances. Instead of using all $N$ views at once, a more efficient approach is to first examine one view and then select another view to resolve ambiguities from the initial glance. If the initial side view cannot distinguish between a guitar and a mandolin (a round-shaped instrument that may also have a flat back), we can then query the front view.

following Mnih et al. (2014), we train the controller via *trial-and-error*. This process is formulated as a reinforcement learning problem aiming to maximize the final recognition or detection accuracy.

On both multiview classification and detection tasks, our experimental results show that MVSelect can provide a good strategy for fixed task networks, while also capable of joint training with the task network for further performance improvements. Specifically, when joint training both MVSelect and the task network, the resulting system can achieve competitive performance to using all $N$ cameras, while using only 2 views for classification tasks and 3 views for detection tasks, respectively.

The computational overhead of MVSelect is very small, as it shares the feature extraction backbone with the task network and only has a few learnable layers. For the entire system, the computational cost is roughly proportional to the number of views used, *e.g.*, approximately $2/N$ of the total computation when 2 views are used, a significant efficiency boost.

The MVSelect policy also enables study on multiview camera layout. In fact, we find that many of the $N$ cameras are rarely chosen and can be shut off for further operational cost improvements, which can serve as a starting point for future study on multiview camera layout optimization.

## 2 BACKGROUND

**Multiview classification.** One effective way for 3D shape recognition is to capture the object in multiple camera views. MVCNN (Su et al., 2015) extracts feature vectors from the input views, and then uses max pooling to aggregate across multiple views for classification. Based on MVCNN, many alternative approaches are proposed. Qi et al. (2016) propose sphere rendering at different volume resolutions. GVCNN (Feng et al., 2018) investigates hierarchical information between different views by grouping the image features before the final aggregation. RotationNet (Kanezaki et al., 2018) introduces a multi-task objective by jointly considering classification and camera poses. ViewGCN (Wei et al., 2020) uses a Graph Convolution Network (GCN) (Kipf & Welling, 2016) instead of the max pooling layer to aggregate across views. Recently, Hamdi et al. (2021) propose the MVTN network to estimate the best viewpoints for 3D point cloud models.

**Multiview detection.** Occlusion is a key issue for object detection using only one camera view. To deal with this problem, researchers investigate multiview approaches for pedestrian detection and estimate occupancy from the bird's-eye-view (BEV). For this task, some methods (Fleuret et al., 2007; Roig et al., 2011; Xu et al., 2016) aggregate single-view detection results. Others find single-view detection results unreliable and instead aggregate the features. Hou et al. (2020) introduce MVDet, the first fully deep-learning approach, which projects feature maps from each camera view to the BEV. Based on MVDet, researchers develop other deep methods. SHOT (Song et al., 2021)

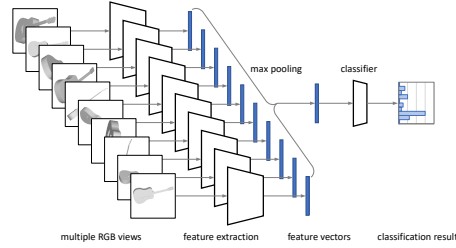 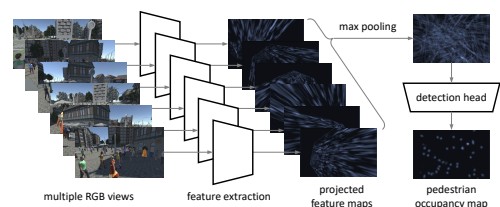

Figure 3: Multiview classification with MVCNN (Su et al., 2015).

Figure 4: Multiview detection with MVDet (Hou et al., 2020).

projects the image feature map at different heights and stacks them together to improve performance. MVDeTr (Hou & Zheng, 2021) deals with distinct distortion patterns from the projection. Qiu et al. (2022) investigate data augmentation with simulated occlusion over multiple views.

**Camera viewpoint study.** For reconstruction with structure-from-motion, Cao & Snavely (2014) finds a representative set of 3D point clouds. Active vision community investigates next-best-view planning with *3D sensors* like RGBD cameras or LiDAR for scene reconstruction (Scott et al., 2003; Chen et al., 2011; Border et al., 2018). In multiview classification, RotationNet (Kanezaki et al., 2018) makes some pioneering work on limited view numbers by taking a random partial set of all $N$ views of the image. MVTN (Hamdi et al., 2021) uses the 3D point cloud as initial input and then estimates the best camera layout for multiview classification, but its moving camera assumption is hard to met in real-world systems. In multiview detection, Vora et al. (2021) investigate camera layout generalization by randomly dropping camera views from both training and testing. For 3D human pose estimation, Pirinen et al. (2019) actively select cameras over a dome.

**Reinforcement learning** (RL) directs an agent to interact with the environment in a manner that maximizes cumulative rewards. State $s \in \mathcal{S}$, action $a \in \mathcal{A}$, and reward $r \in \mathcal{R}$ are key concepts to model the interaction between agent and environment. In a certain state $s$, policy $\pi(a|s)$ records the probability for each action, and state value function $V(s)$ estimates the future rewards when following the corresponding policy. To learn the best policy, Q-learning and DQN (Mnih et al., 2013) optimize the action value function $Q(s, a)$, which describes the estimated future return for a specific action $a$ at state $s$. Policy gradient methods like REINFORCE (Williams, 1992) and PPO (Schulman et al., 2017) directly optimize for the polity $\pi(a|s)$.

## 3 MULTIVIEW NETWORK REVISIT

### 3.1 MULTIVIEW CLASSIFICATION WITH MVCNN

MVCNN (Su et al., 2015) (Fig. 3) is a classic architecture which many multiview classification networks build upon. Given $N$ input images $\boldsymbol{x}_n, n \in \{1, \dots, N\}$, first, MVCNN uses its feature extractor $f(\cdot)$ to calculate the feature vectors,

$$\boldsymbol{h}_n = f(\boldsymbol{x}_n), \tag{1}$$

where the feature vector $\boldsymbol{h}_n \in \mathbb{R}^D$ is $D$-dimensional. Secondly, it uses max pooling to aggregate multiple views into an overall feature descriptor $\hat{\boldsymbol{h}} \in \mathbb{R}^D$,

$$\hat{\boldsymbol{h}} = \max_n \{\boldsymbol{h}_n\}, \tag{2}$$

where $\max\{\cdots\}$ takes the maximum along each of the $D$ dimensions. Lastly, it applies the output head $g(\cdot)$ to produce the classification result $\hat{\boldsymbol{y}}$,

$$\hat{\boldsymbol{y}} = g\left(\hat{\boldsymbol{h}}\right). \tag{3}$$

In training, the original design by Su et al. (2015) adopts a 2-stage paradigm by first training on individual views and then considering multiple views. In this paper, we skip the first stage and directly train MVCNN on all $N$ views,

$$\mathcal{L}_{\text{MVCNN}} = \mathcal{L}_{\text{CE}}(\hat{\boldsymbol{y}}, \boldsymbol{y}), \tag{4}$$

where $\mathcal{L}_{\text{CE}}(\cdot, \cdot)$ denotes the cross-entropy loss and $\boldsymbol{y}$ denotes the ground truth one-hot label.

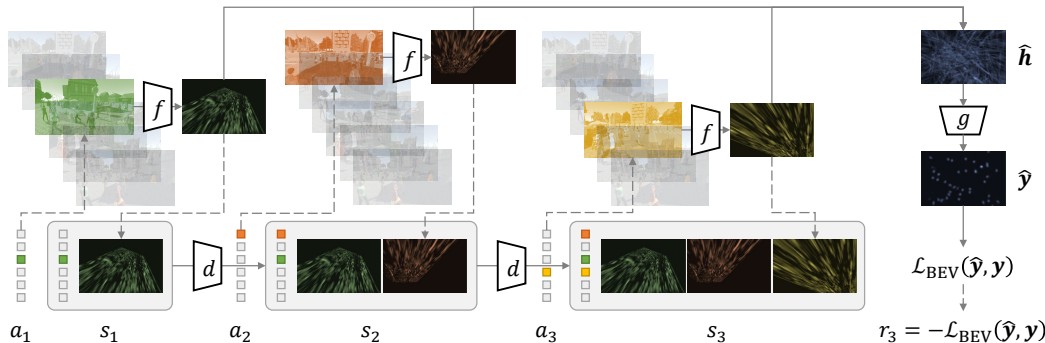

Figure 5: Efficient multiview understanding with $T = 3$ glances. Solid lines indicate the network forward pass and dashed lines indicate the interaction between *agent* (MVSelect) and *environment* (multiview system). The approach starts with a random initial view $a_1$, and the feature extractor $f(\cdot)$ computes its feature (Eq. 1). The *state* for this initial time step is recorded as $s_1$ (Eq. 6). Next, the proposed MVSelect module $d(\cdot)$ is used to choose a second view $a_2$, and the *state* is updated as $s_2$. Then, by repeating the last step, a third view $a_3$ is chosen and the state is updated as $s_3$. Finally, three views are aggregated into an overall descriptor $\hat{h}$ (Eq. 2), and the task network output $\hat{y}$ is calculated using the output head $g(\cdot)$ (Eq. 3). The final *reward* $r_3 = \text{MODA}(\hat{y}, y)$ is set as the accuracy fir the task network (Eq. 7), and all other rewards are set as zero.

## 3.2 MULTIVIEW DETECTION WITH MVDET

MVDet (Hou et al., 2020) (Fig. 4) is a multiview detection architecture that is followed by many recent works. To estimate human occupancy in bird's-eye-view (BEV), given input images $x_n, n \in \{1, \ldots, N\}$, MVDet first extracts $D$-channel feature maps for each view and uses perspective transformation to project the camera views to the BEV. Together, these operations can be considered as the BEV feature extraction step under Eq. 1, with the exception that $h_n \in \mathbb{R}^{D \times H \times W}$ now denotes the $D$-channel feature map for the BEV scenario of shape $H \times W$. Secondly, for multiview aggregation, instead of the concatenation in the original design, we choose element-wise max pooling in Eq. 2, producing the overall feature description $\hat{h} \in \mathbb{R}^{D \times H \times W}$ that fits arbitrary numbers of views. Lastly, we apply the output head as Eq. 3 to generate a heatmap $\hat{y} \in (0, 1)^{H \times W}$ that indicates the likelihood of human occupancy in each BEV location. The loss for MVDet can be written as,

$$\mathcal{L}_{\text{MVDet}} = \mathcal{L}_{\text{BEV}}(\hat{y}, y) + \frac{1}{N} \sum_{n=1}^{N} \mathcal{L}_n, \tag{5}$$

where $\mathcal{L}_{\text{BEV}}(\cdot, \cdot)$ denotes BEV output loss; $y \in \{0, 1\}^{H \times W}$ denotes binary ground truth map; and $\mathcal{L}_n$ denotes the auxiliary per-view loss with 2D bounding boxes.

## 4 EFFICIENT MULTIVIEW UNDERSTANDING

In a multiview system with $N$ views, our approach uses a total of $T < N$ camera views $a_t \in \{1, \ldots, N\}, t \in \{1, \ldots, T\}$ for efficient understanding. To achieve this, we propose a view selection module, MVSelect, denoted as $d(\cdot)$, that sequentially selects camera views. Starting from a random initial view $a_1$, MVSelect chooses the remaining $T - 1$ cameras by observing the target object or scene from existing views $a_1, \ldots, a_t$ at each time step $t$ and deciding which camera view $a_{t+1}$ to select next. The resulting $T$ views should give the task network high classification or detection performance. Once $T$ camera views have been gathered, we aggregate them into an overall description $\hat{h}$ using Eq. 2, and calculate the final output $\hat{y}$ using Eq. 3.

Fig. 5 gives an overview of the proposed efficient multiview approach.

---

**Algorithm 1** Joint training of MVSelect and task network.

---

1: **input**: camera views $\boldsymbol{x}_n, n \in \{1, \ldots, N\}$, ground truth $\boldsymbol{y}$, random initial view $a_1$, number of total views $T$, hyper-parameter $\epsilon$.
2: **update**: feature extractor $f(\cdot)$ and output head $g(\cdot)$ of task network, and MVSelect controller $d(\cdot)$.
3: initialize the total loss $\mathcal{L}_{\text{total}} = 0$;
4: **for** $t \in \{1, \ldots, T-1\}$ **do**
5:      select and apply the next action using $\epsilon$-greedy: with probability $\epsilon$ adopt a random action, or else choose the action with highest value $a_{t+1} = \arg\max_a Q(s_t, a)$;
6:      observe next state $s_{t+1}$ (Eq. 6) and reward $r_{t+1}$ (Eq. 7);
7:      calculate the RL loss $\mathcal{L}_{\text{RL}}$ (Eq. 9) and update the total loss $\mathcal{L}_{\text{total}} = \mathcal{L}_{\text{total}} + \mathcal{L}_{\text{RL}}$;
8: **end for**
9: calculate the task loss $\mathcal{L}_{\text{task}}$ (Eq. 4 or Eq. 5) and update the total loss $\mathcal{L}_{\text{total}} = \mathcal{L}_{\text{total}} + \mathcal{L}_{\text{task}}$;
10: optimize for the total loss $\mathcal{L}_{\text{total}}$.

---

## 4.1 PROBLEM FORMULATION

While iteratively selecting camera views, it is impossible to know what is in the *not-selected* camera views. Those views cannot propagate gradients back to the controller, making the problem *non-differentiable*. Therefore, the controller has to learn through *trial and error*, and we formulate this non-differentiable process as a reinforcement learning problem, where MVSelect is the *agent* and the multiview system is the *environment*.

**State.** In order to get a Markovian representation, we record the chosen camera views $a_1, \ldots, a_t$ and the observations $\boldsymbol{h}_{a_1}, \ldots, \boldsymbol{h}_{a_t}$ as state $s_t$. We use the extracted features to represent the observations rather than the RGB camera views to reduce dimensionality and maximize efficiency, since these features will be used in the task network later (Eq. 2). Mathematically, we formulate the state $s_t$ as,

$$s_t = \left\langle s_t^{\text{cam}}, s_t^{\text{obs}} \right\rangle,$$

$$s_t^{\text{cam}} = \sum_{\tau=1}^{t} \text{onehot}(a_\tau), \tag{6}$$

$$s_t^{\text{obs}} = \max_{\tau=1}^{t} \{\boldsymbol{h}_{a_\tau}\},$$

where $\text{onehot}(\cdot)$ is the one-hot function over $N$ cameras. This representation reflects both chosen cameras $s_t^{\text{cam}} \in \mathbb{R}^N$ and their observations $s_t^{\text{obs}} \in \mathbb{R}^D$, and maintains the same dimensionality across different time steps. The observation part $s_t^{\text{obs}}$ also matches the overall representation in Eq. 2.

**Action.** For state $s_t, t \in \{1, \ldots, T-1\}$, MVSelect takes the next camera view $a_{t+1}$ as action.

**Reward.** Upon taking action $a_{t+1}$, the system receives reward $r_{t+1}$ and transitions into the next state $s_{t+1}$. To achieve high task network performance, we consider the following as reward,

$$r_t = 0, t \in \{1, \ldots, T-1\},$$
$$r_T^{\text{MVCNN}} = \mathbb{1}(\hat{\boldsymbol{y}} = \boldsymbol{y}), \; r_T^{\text{MVDet}} = \text{MODA}(\hat{\boldsymbol{y}}, \boldsymbol{y}), \tag{7}$$

where $\mathbb{1}(\cdot)$ denotes the binary indicator function, $\text{MODA}(\cdot, \cdot)$ is the evaluation metric for multiview detection (Kasturi et al., 2008). We also experiment with other reward designs in Section 5.4.

## 4.2 MVSELECT ARCHITECTURE

We design MVSelect architecture $d(\cdot)$ with two branches. The first branch expands the camera selection result $s_t^{\text{cam}} \in \mathbb{R}^N$ into $D$-dimensional learnable camera embeddings, and then sums over the selected embeddings to formulate a hidden vector. The second branch processes the observation $s_t^{\text{obs}} \in \mathbb{R}^D$, and converts that into another hidden vector. By combining the two hidden vectors, the controller network outputs the action-value $Q(s, a)$, which measures the expected cumulative rewards for taking an action $a$ in a given state $s$. Please see Appendix A.1 for figure illustrations.

During testing, MVSelect outputs the next action as,

$$a_{t+1} = \arg\max_a Q(s_t, a),$$

which maximizes the expected cumulative rewards.

### 4.3 TRAINING SCHEME

We adopt Q-learning (Sutton & Barto, 2018) for training MVSelect. Specifically, action-value function $Q(\cdot, \cdot)$ should estimate the cumulative future rewards after taking action $a_{t+1}$ at state $s_t$,

$$Q(s_t, a_{t+1}) = \mathbb{E}\left(\sum_{\tau=t+1}^{T} \gamma^{\tau-t-1} r_\tau\right),$$

where $\mathbb{E}(\cdot)$ denotes the expectation, and $\gamma \in [0,1]$ denotes the discount factor. We take the temporal difference (TD) (Sutton & Barto, 2018) target as supervision for the action value,

$$q_t = \begin{cases} r_{t+1} + \gamma \max_a Q(s_{t+1}, a), & \text{if } t < T-1 \\ r_T, & \text{otherwise} \end{cases}, \tag{8}$$

and calculate the loss using the $L_2$ distance,

$$\mathcal{L}_{\text{RL}} = \sum_{t=1}^{T-1} \mathcal{L}_{\text{MSE}}(Q(s_t, a_{t+1}), q_t), \tag{9}$$

where the next action $a_{t+1}$ is chosen using $\epsilon$-greedy for exploration-exploitation trade offs.

In joint training, the task network takes supervision from the task loss $\mathcal{L}_{\text{task}}$ (Eq. 4 and Eq. 5), and the selection module takes supervision from the RL loss $\mathcal{L}_{\text{RL}}$ (Eq. 9).

A step-by-step demonstration of this process can be found in Algorithm 1.

## 5 EXPERIMENTS

### 5.1 EXPERIMENT SETTINGS

We summarize the experiment settings as follows. For more details, please see Appendix A.2

**Datasets.** For multiview classification, we use synthetic dataset ModelNet40 (Wu et al., 2015) under two different camera layouts (12 views and 20 views) and real-world dataset ScanObjectNN (12 views) (Uy et al., 2019). For multiview detection, we use real-world dataset Wildtrack (7 views) (Chavdarova et al., 2018) and synthetic dataset MultiviewX (6 views) (Hou et al., 2020).

**Evaluation metrics.** For multiview classification, we report instance-averaged accuracy. Regarding multiview detection, we report multi-object detection accuracy (MODA), which is calculated as $1 - \frac{\text{FP}+\text{FN}}{\text{GT}}$ (Kasturi et al., 2008). All metrics are reported in percentages.

**Implementation details.** For multiview classification, we input images of size $224 \times 224$ to the MVCNN model. For multiview detection, we use a resolution of $720 \times 1280$ for input images with view-coherent data augmentation (Hou & Zheng, 2021), and downsample the BEV grid by a factor of 4. In terms of architecture, we use ResNet-18 (He et al., 2016) as feature extractor $f(\cdot)$.

We train all networks for 10 epochs using the Adam optimizer (Kingma & Ba, 2015). We use learning rates of $5 \times 10^{-5}$ and $5 \times 10^{-4}$, with batch sizes of 8 and 1 for MVCNN and MVDet, respectively. The MVSelect module is trained using a learning rate of $1 \times 10^{-4}$. For joint training, we decrease the learning rate for the task network to $1/5$ of its original value. Regarding hyperparameters, we set the future reward discount factor $\gamma = 0.99$, and the exploration ratio $\epsilon$ to gradually decrease from 0.95 to 0.05 during training. All experiments are conducted on a single RTX-3090 GPU and averaged across 5 repetitive runs.

**Experimental setups.** First, we fix the task network and only train the view selection module. Our goal is to show that the view selection results outperform three existing view selection methods including 1) random selection, 2) the best policy on validation set, and 3) the policy that maximizes field-of-view (FoV) coverage, which is modified from a next-best-view work by Scott et al. (2003) and therefore only applicable to multiview detection. We also report two oracles: for a certain initial view, 1) choosing the overall best-performing camera for all instances in the dataset (*dataset-level*

Table 1: Evaluation on multiview classification.

| view selection ($T = 2$) | ModelNet40 | | ScanObjectNN |
|---|---|---|---|
| | 12 views | 20 views | |
| N/A: all $N$ views | 94.5 | 96.5 | 86.1 |
| dataset-lvl oracle | 85.2 ± 1.4 | 69.9 ± 1.9 | 78.5 ± 1.2 |
| instance-lvl oracle | 96.5 ± 0.4 | 98.1 ± 1.9 | 93.4 ± 0.6 |
| random selection | 71.5 ± 2.5 | 48.1 ± 3.1 | 74.2 ± 3.5 |
| validation best policy | 85.1 ± 0.8 | 69.5 ± 1.0 | 77.5 ± 1.1 |
| MVSelect | 88.2 ± 0.4 | 79.6 ± 1.8 | 80.0 ± 0.8 |
| MVSelect + joint training | 94.3 ± 0.2 | 94.4 ± 0.2 | 84.1 ± 0.2 |

Table 2: Evaluation on multiview detection.

| view selection ($T = 3$) | Wildtrack | MultiviewX |
|---|---|---|
| N/A: all $N$ views | 90.0 | 93.0 |
| dataset-lvl oracle | 82.5 ± 0.4 | 80.2 ± 0.3 |
| instance-lvl oracle | 87.4 ± 0.4 | 82.3 ± 0.8 |
| random selection | 74.9 ± 1.3 | 76.2 ± 1.4 |
| validation best policy | 79.5 ± 1.1 | 78.0 ± 0.4 |
| max FoV (Scott et al., 2003) | 78.0 ± 1.5 | 73.9 ± 1.5 |
| MVSelect | 80.0 ± 0.8 | 78.7 ± 0.5 |
| MVSelect + joint training | 88.6 ± 0.2 | 88.1 ± 0.2 |

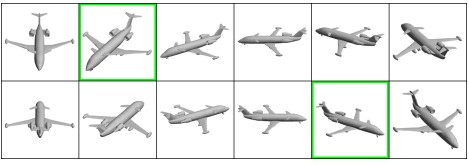 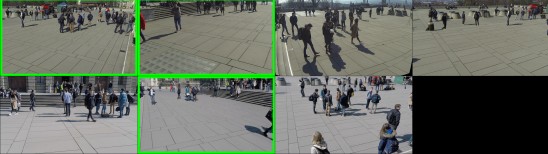

Figure 6: Example of selected views on ModelNet40 (left) and Wildtrack (right).

*oracle*) and 2) choosing specifically for each instance (*instance-level oracle*). The former reflects the upper bound of performance achievable by human-designed heuristics, and the latter represents the theoretical performance ceiling when keeping the task network fixed.

Second, we jointly train the proposed view selection module with the task networks. For this experiment, our goal is to achieve the highest possible performance using $T$ views. If not specified, we use a total of $T = 2$ views for multiview classification and $T = 3$ views for detection.

## 5.2 EVALUATION OF MVSELECT

In this section, we compare our efficient approach against existing baselines for view selection. For comparisons between our efficient approach and state-of-the-art methods, please see Appendix A.3.

For **multiview classification**, as shown in Table 1, randomly selecting two views cannot achieve competitive results. The greedy policy learned on the validation set gives better results that are more in line with the dataset-level oracle.

The proposed view selection module, on the other hand, can choose the supplementary view very effectively. On ModelNet40 dataset (Wu et al., 2015), MVSelect with fixed MVCNN outperforms the random selection baseline by large margins on both settings of ModelNet40 and on ScanObjectNN. Compared to dataset-level oracles (same policy for all instances with the same initial camera), MVSelect also turns out to be advantageous by 3.0%, 9.7%, and 1.5% across the two settings. This verifies that MVSelect can take the target object into consideration (see Fig. 2) and select different cameras for different instances under the same initial camera.

When joint training MVCNN with MVSelect, we witness large improvements compared to keeping MVCNN fixed. In fact, on two settings, the results are only 0.2%, 2.1%, and 2.0% behind compared to the full $N$-view system. Overall, we believe that MVSelect and its joint training capabilities enable us to consider only 2 views without major performance drawbacks. We demonstrate an example of the selected views in Fig. 6, and the MVSelect policy in Fig. 7.

For **multiview detection**, we report view selection results in Table 2. Since the target scenarios are not fully captured by any individual camera, randomly selecting $T = 3$ cameras does not yield satisfactory results. The best policy found on validation set gives better performance compared to random selection. However, the maximum FoV coverage policy (Scott et al., 2003) gives mixed results, suggesting that FoV coverage as a heuristic cannot guarantee the best detection result, possibly due to heavy occlusions. In addition, we observe that the instance-level oracle remains relatively low compared to that of multiview classification tasks (see Table 1). This is likely due to the target scenarios not being fully captured by any single view, and the multiview detection network needs multiple views to collaborate with each other for optimal results.

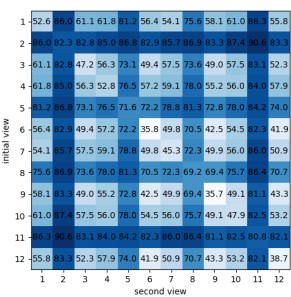


(a) Classification accuracy      (b) MVSelect policy

Figure 7: Multiview classification with $T = 2$ views on the 12-view setup of ModelNet40. The task network is fixed once trained. **Left**: test set accuracy of using two views. **Right**: MVSelect policy for the test set.

Table 3: Computation efficiency.

|  | views | FLOPs | | | throughput (instance/s) |
|---|---|---|---|---|---|
|  |  | $f(\cdot)$ | $g(\cdot)$ | $d(\cdot)$ |  |
| ModelNet40 | 20 | 36.5G | 20.5k | N/A | 119.6 |
|  | 2 | 3.6G | 20.5k | 1.1M | 361.8 |
|  | 12 | 21.9G | 20.5k | N/A | 196.2 |
|  | 2 | 3.6G | 20.5k | 530.4k | 507.9 |
| Wildtrack | 7 | 1.2T | 19.1G | N/A | 8.4 |
|  | 3 | 511.6G | 19.1G | 3.2G | 16.2 |
| MultiviewX | 6 | 1.0T | 17.7G | N/A | 9.8 |
|  | 3 | 511.6G | 17.7G | 2.9G | 16.9 |

Table 4: Ablation study.

|  | ModelNet40 12 views | Wildtrack |
|---|---|---|
| MVSelect | 88.2 | 80.0 |
| w/o camera branch | 88.2 | 79.1 |
| w/o feature branch | 85.0 | 79.7 |
| w/ transformer | 87.0 | 78.7 |
| reward=$\Delta$ task loss | 88.0 | 80.1 |

In this scenario, we find that MVSelect with a fixed task network can outperform existing baselines, including random selection, greedy policy on validation, and maximum FoV coverage (Scott et al., 2003). Although the raw improvements are not as substantial as those in multiview classification, they are **statistically highly significant** (p-value $< 0.001$). Compared to the dataset-level oracles, we find the MVSelect policy lose its edge. In fact, we find that for multiview detection, MVSelect tends to select the same camera for a given initial view, since it is *not* aware of the situation outside of the FoV coverage. As a result, it cannot choose cameras based on *uncovered* areas in different frames. Without this instance-aware advantage, the fixed camera policy learned during training (MVSelect) cannot outperform the dataset-level oracle, whose policy is computed on the test set.

Joint training with MVDet once again leads to substantial performance improvements over keeping the task network fixed. In fact, the results even exceed the instance-level oracle for fixed task networks. Using $T = 3$ views, the joint training approach provides competitive results to using all $N$ views, and exceeds the reported performance in the original MVDet paper (Hou et al., 2020). We present an example of the selected views in the Wildtrack dataset in Fig. 6.

### 5.3 EFFICIENCY ANALYSIS

In Table 3, following previous works in efficient inference (Li et al., 2017; Howard et al., 2017), we detail the computational cost in FLOPs for task networks and MVSelect. Specifically, we find feature extraction $f(\cdot)$ to take up the majority of the computation, while everything else is lighter by at least an order of magnitude. Overall, we verify that using $T = 2$ or 3 out of $N$ views can reduce the computational cost to roughly $T/N$.

In terms of inference speed, we find reduction in FLOPs results in monotonically increasing throughputs, ranging from $1.72\times$ to $3.03\times$. Due to factors such as implementation, parallelization, and hardware limitations, actual speedups cannot actually reach the level of computational cost reduction, as suggested by previous study (Molchanov et al., 2016).

### 5.4 VARIANT STUDY

**Ablation study.** Regarding the MVSelect architecture design, Table 4 shows that removing the camera branch and feature branch primarily affects multiview detection and multiview classification performance, respectively. This aligns with the policy we learned for the two tasks. For multiview detection, since the system has no clue about areas outside camera FoVs, the camera branch plays a more important role for encoding prior knowledge of the scene layout. For multiview classification, however, target objects are fully observable, and MVSelect can make different decisions for each instance. Thus, the feature branch is more important, as it enables per-instance decision making.

When changing the MVSelect network architecture into transformer (Vaswani et al., 2017), we find that more parameters do not translate into better performance.

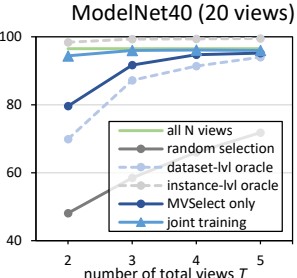 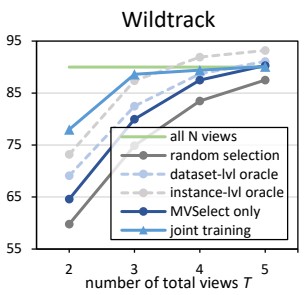 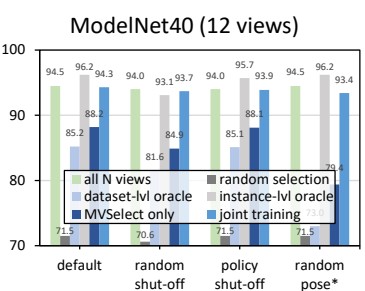

Figure 9: System performance under different number of total views $T$. We report classification accuracy and MODA for the two tasks, respectively (same for Fig. 10 and Table. 4). Circle and triangle markers indicate whether the task network is fixed or not, respectively. Dotted lines represent oracle performance (not achievable).

Figure 10: System evaluation under different settings. Except for the random pose* setting (where MVSelect variants are re-trained), all models are trained on the default setting.

Another option for the reward is the change in task loss. In our variant study, this reward design does not show any significant advantages over the current design.

**Influence of total view count.** The performance curves in Fig. 9 demonstrate that the joint training variant achieves competitive performance with as few as $T = 2$ views for multiview classification and $T = 3$ views for multiview detection, beyond which point performance plateaus. By contrast, learning MVSelect for fixed task networks shows a less steep curve. Notably, both datasets exhibit an increase in performance up to a total of $T = 5$ views, at which point the MVSelect policy performs comparably to the full $N$-view system.

**Camera layout optimization.** Determining the optimal camera locations is crucial for setting up an effective multiview system. In Fig.7, we observe that not all cameras are equally useful according to the MVSelect policy. To address this, we allocate a validation partition of the data to identify the more useful camera views and then disable half of the $N = 12$ cameras that are not frequently utilized. In testing, we find that this new camera layout (policy shut-off" in Fig. 10) does not significantly impact the task network or the MVSelect module, and outperforms randomly disabling 6 of the 12 cameras (random shut-off" in Fig.10). Although it is necessary to set up all $N$ cameras for the analysis, optimizing the multiview camera layout can be a crucial step towards achieving optimal performance, and merits further investigation.

**Random object pose.** In real-world applications of multiview systems, such as those found in iPhones and Teslas, the cameras may remain fixed in their relative positions while the entire system is in motion. To simulate this scenario in our experiments, we introduce the random object pose setting (Fig. 10) and re-train MVSelect. While there is no exact object pose as supervision, the reinforcement learning approach is able to roughly infer the relative poses between the object and the multiview system, resulting in improved performance compared to random selection and dataset-level oracle (which is arguably inappropriate for this setup). In the future, we plan to estimate camera poses with respect to the object or the environment as an auxiliary supervision for moving setups.

## 6  CONCLUSION

In conclusion, this paper proposes an efficient approach for multiview understanding by limiting the number of views. To this end, a camera view selection module, MVSelect, is proposed along with a reinforcement learning based training scheme that can learn from the non-differentiable selection process. When jointly trained with the task network, the proposed approach demonstrates competitive performance on multiview classification and detection tasks at fractions of the computational cost. Overall, the proposed efficient approach provides an alternative to reducing image resolution and using lighter networks, and paves ways for future multiview camera layout optimization.

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

# A   APPENDIX

## A.1   MVSELECT ARCHITECTURE

As shown in Fig. 11, we design MVSelect architecture $d\left(\cdot\right)$ with two branches. The first branch expands the camera selection result $s_t^{\text{cam}} \in \mathbb{R}^N$ into $D$-dimensional learnable camera embeddings, and then sums over the selected embeddings to formulate a hidden vector. The second branch processes the observation $s_t^{\text{obs}} \in \mathbb{R}^D$, and converts that into another hidden vector. By combining the two hidden vectors, MVSelect outputs the action-value $Q\left(s, a\right)$, which measures the expected cumulative rewards for taking an action $a$ in a given state $s$.

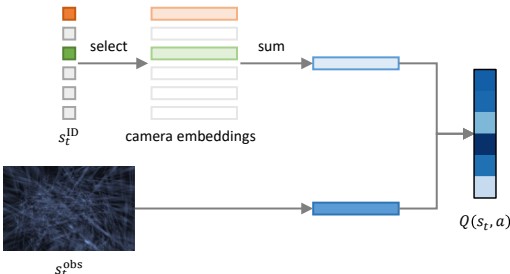

Figure 11: MVSelect architecture.

## A.2   ADDITIONAL DETAILS ON EXPERIMENTAL SETUP

**Datasets.** We verify the performance of the proposed approach on multiview classification and detection tasks.

*ModelNet40* is a subset of 3D CAD models in ModelNet Wu et al. (2015). It includes 40 categories of synthetic 3D objects with 9,843 training models and 2,468 test models. For multiview classification experiments, we use two different configurations: the *12-view* circular configuration from MVCNN Su et al. (2015) and the *20-view* dodecahedral configuration from RotationNet Kanezaki et al. (2018).

*ScanObjectNN* is a 3D dataset scanned from real-world objects. Introduced by Uy et al. (2019), it contains 2902 3D objects across 15 categories. Traditionally used for point cloud classification, we re-purpose this dataset for multiview classification by rendering textured meshes from the point clouds and use the same 12 views setup as ModelNet40 (Wu et al., 2015; Su et al., 2015).

*Wildtrack* Chavdarova et al. (2018) is a real-world multiview detection dataset with 7 camera views covering a $12 \times 36$ square meter area, which is represented as a $480 \times 1440$ grid from BEV. It contains 360 frames for training and 40 frames for testing.

*MultiviewX* Hou et al. (2020) is a synthetic multiview detection dataset created using the Unity Technologies engine. It has 6 cameras with higher pedestrian density than Wildtrack. It focuses on a $16 \times 25$ square meter area, which is discretized into $640 \times 1000$ BEV grid. Like Wildtrack, MultiviewX also contains 360 training frames and 40 testing frames.

**Evaluation metrics.** For multiview classification, we follow previous methods Qi et al. (2016); Kanezaki et al. (2018); Wei et al. (2020); Yu et al. (2018); Yang & Wang (2019); Hamdi et al. (2021) and report instance-averaged accuracy as the primary indicator.

Regarding multiview detection, we report the following metrics: multi-object detection accuracy (MODA), multi-object detection precision (MODP), precision, and recall Kasturi et al. (2008). During evaluation, we first compute false positives (FP), false negatives (FN), and true positives (TP), and then use them to calculate the metrics. Specifically, MODA is calculated as $1 - \frac{\text{FP}+\text{FN}}{\text{GT}}$, where GT is the number of ground truth pedestrians. MODP is calculated as $\frac{\sum 1-\text{dist}[\text{dist}<\text{thres}]/\text{thres}}{\text{TP}}$, where dist is the distance from the estimated pedestrian location to its ground truth and thres is the threshold of 0.5 meters. MODP indicates the BEV localization accuracy. Precision and recall are calculated as $\frac{\text{TP}}{\text{TP}+\text{FP}}$ and $\frac{\text{TP}}{\text{GT}}$, respectively.

Table 5: Performance comparison with state-of-the-art multiview classification and multiview detection methods. Results are averaged from 5 runs. * indicates that the camera poses are dynamically chosen and do not follow a pre-defined layout. We also report the MVSelect and task network joint training results in the last line.

| | ModelNet40 Wu et al. (2015) | |
|---|---|---|
| | 12 views | 20 views |
| MVCNN Su et al. (2015) | 90.1 | 92.0 |
| GVCNN Feng et al. (2018) | 92.6 | - |
| MHBN Yu et al. (2018) | 93.4 | - |
| RotationNet Kanezaki et al. (2018) | - | 94.7 |
| RelationNet Yang & Wang (2019) | 94.3 | 97.3 |
| ViewGCN Wei et al. (2020) | - | **97.6** |
| MVTN* Hamdi et al. (2021) | 93.8 | 93.5 |
| MVCNN (our implementation) | **94.5** | 96.5 |
| MVCNN + MVSelect (2 views) | 94.3 | 94.4 |

| | Wildtrack Chavdarova et al. (2018) | | | | MultiviewX Hou et al. (2020) | | | |
|---|---|---|---|---|---|---|---|---|
| | MODA | MODP | prec. | recall | MODA | MODP | prec. | recall |
| RCNN & cluster Xu et al. (2016) | 11.3 | 18.4 | 68 | 43 | 18.7 | 46.4 | 63.5 | 43.9 |
| POM-CNN Fleuret et al. (2007) | 23.2 | 30.5 | 75 | 55 | - | - | - | - |
| DeepMCD Chavdarova et al. (2017) | 67.8 | 64.2 | 85 | 82 | 70 | 73 | 85.7 | 83.3 |
| Deep-Occlusion Baqué et al. (2017) | 74.1 | 53.8 | 95 | 80 | 75.2 | 54.7 | 97.8 | 80.2 |
| MVDet Hou et al. (2020) | 88.2 | 75.7 | 94.7 | 93.6 | 83.9 | 79.6 | 96.8 | 86.7 |
| SHOT Song et al. (2021) | 90.2 | 76.5 | 96.1 | 94.0 | 88.3 | 82.0 | 96.6 | 91.5 |
| MVDeTr Hou & Zheng (2021) | **91.5** | **82.1** | **97.4** | 94.0 | **93.7** | **91.3** | **99.5** | 94.2 |
| MVDet (our implementation) | 90.0 | 80.9 | 95.4 | **94.5** | 93.0 | 90.3 | 98.7 | **94.4** |
| MVDet + MVSelect (3 views) | 88.6 | 79.9 | 93.3 | 94.2 | 88.1 | 89.8 | 98.2 | 89.7 |

All metrics are reported in percentages.

## A.3 EVALUATION AGAINST STATE-OF-THE-ARTS

In Table 5, we compare our implementations of MVCNN Su et al. (2015) and MVDet Hou et al. (2020) with their original implementations and state-of-the-art methods. On 3 datasets and 4 settings, our implementations outperform the original implementations and achieve competitive results. Although our focus is not on improving these classic architectures, the results indicate that they can still serve as strong baselines.

Compared to state-of-the-arts that use full $N$ cameras, joint training the tasks network along with MVSelect gives competitive results while only using $T = 2$ or $T = 3$ cameras for multiview classification and multiview detection.

