# OpenReview forum: "Learning to Select Camera Views: Efficient Multiview Understanding at Few Glances"
_ICLR.cc/2024/Conference — ICLR 2024 Conference Withdrawn Submission_

### Official Review · Reviewer_SBYb · 2023-10-30

**Soundness:** 2 fair
**Presentation:** 3 good
**Contribution:** 3 good
**Rating:** 5
**Confidence:** 4

**Summary:**

The paper proposes a reinforcement learning based camera view selection approach, called MVSelect, joint training with the task network. Experimental results on multiview classification and detection tasks show that the approach achieves promising performance.

**Strengths:**

A reinforcement learning based camera view selection approach. The proposed method is evaluated on multi-view classification and multi-view detection.

**Weaknesses:**

Weakness:
1.	Literature review:
3D reconstruction, multi-view counting or tracking is also related to the proposed multi-view selection task. They should also be reviewed in the paper.
Camera selection for 3D reconstruction is a key step. There’re a lot of camera view selection research in the area of 3D reconstruction. They should be well reviewed in the paper. To just name a few:
[1] Learning Reconstructability for Drone Aerial Path Planning, SIGGRAPH 2022.
[2] Offsite Aerial Path Planning for Efficient Urban Scene Reconstruction, SIGGRAPH ASIA 2020.
There are also a lot camera view fusion and selection based methods in other multi-camera surveillance, such as “Cross-view cross-scene multi-view crowd counting”, CVPR 2021.

2.	I highly doubt about the generality of the method. Can the proposed method training on one scene be applied to any other new scenes? If not, the method’s application scenarios are very limited.
3.	Why use MODA as reward? Why not use task’s loss value as reward?
4.	There are no reasonable baselines methods at all. Actually, some simple constraint on the camera views may be already effective. For example, in the multi-view detection, if we could make sure the selected camera views can cover most of the scene areas, the performance can be also very good.
5.	We should compare with camera selection methods in the 3D reconstruction as well.
6.	We should compare with DNNs-based camera view selection modules as well.
7.	The paper emphasizes that the computation cost is very small. But there is no computation cost or running speed comparison with other camera selection methods at all.
8.	Why only use MODA as a metric in multi-view detection? MODA, MODP, Precision, Recall should all be used in the evaluation.
9.	Is there an ablation study on the selected camera view number?

**Questions:**

As above.

---

### Official Review · Reviewer_xRSL · 2023-10-31

**Soundness:** 3 good
**Presentation:** 3 good
**Contribution:** 2 fair
**Rating:** 5
**Confidence:** 3

**Summary:**

In this paper, the authors propose a view selection module MVSelect that learns to choose only the best T views for multiview classification and detection instead of using all N camera views (T<<N). Starting with a random initial view, the proposed method sequentially decides the best view based on past camera selections and image features. To train the module, the authors formulate it as a reinforcement learning problem and adopt Q-learning to learn the action-value function. Experiments show that the proposed selection policy outperforms other heuristic policies in multiple datasets for multiview classification (T=2) and detection (T=3). After joint training with the task network, the proposed method is able to achieve similar performance with the full-view baseline, with a reduction in FLOPS and a 1.72x to 3.03x increase in inference speed.

**Strengths:**

1) Though similar selection modules have been proposed in other areas such as efficient video classification and efficient multi-modal learning, I think this paper is novel as it is the first to combine such selection modules with multiview problems.

2) The authors conduct extensive experiments and the proposed method seems to be convincing

**Weaknesses:**

1) Though the proposed method could save a significant amount of FLOPS, the improvement in inference speed is limited as the proposed method is sequential and cannot be paralleled, as shown in Table 3. If the full-view baseline is fully paralleled in real-world applications, the proposed method may even have a lower inference speed.

2) This paper lacks experiments to show the distribution of the selections made by the learned policy. Maybe there are some particular camera views that always outperform the others. Then the learned policy may just select those views blindly without understanding the object geometrics.

3) The clarity of this paper should be improved. For example, in equation (5), there is no definition of the BEV loss and auxiliary per-view loss.

**Questions:**

1) For section 5.3, could the authors provide more details about the calculation throughput? What are the throughputs if T is larger than 2 or 3?

---

### Official Review · Reviewer_cq5m · 2023-11-01

**Soundness:** 2 fair
**Presentation:** 2 fair
**Contribution:** 2 fair
**Rating:** 3
**Confidence:** 5

**Summary:**

Multiview camera setups have proven valuable in computer vision applications by resolving ambiguities, handling occlusions, and expanding field-of-view coverage. However, the computational demands of multiple views pose challenges for resource-constrained devices. To address this, we propose a view selection method that assesses the target subject from available views and selects the most suitable one for recognition or detection. Our approach incorporates an MVSelect1 module, based on reinforcement learning, which not only aids view selection but also enables joint training with the task network. Experimental results in multiview classification and detection tasks demonstrate that our approach achieves promising performance using only a subset of available views (2 or 3 out of N), significantly reducing computational costs. The analysis also suggests potential camera optimization for future multiview systems.

**Strengths:**

Using reinforcement learning frameworks for camera view selection is an interesting idea.

**Weaknesses:**

1.	For MVDter, they use attention for multi-view information aggregation so MVDetr is also including the camera selection concept with better performance than the MVSelect.
2.	Does the ordering of camera selection matter? According to the definition of action in MVSelect, the ordering of camera selection seems like a tunable parameter.

**Questions:**

1.	Can you show some examples of why some view is not selected? Have you analyzed what kind of view is worse than others?
2.	Can you show some failure cases of MVSelect? Does MVSelect make mistakes?

**Details Of Ethics Concerns:**

No concerns.

---

### Official Review · Reviewer_8HDd · 2023-11-02

**Soundness:** 3 good
**Presentation:** 3 good
**Contribution:** 2 fair
**Rating:** 5
**Confidence:** 3

**Summary:**

Paper proposes a method called MVSelect to choose camera views in multi-view classification and multi-view detection settings. The idea is to choose camera which can recover most of the underlying performance and help reducing the computational load. The paper sets its experiments on the basis of MVCNN and MVDet architectures. The paper performance experiments on two classification (ModelNet40, ScanObjectCNN) and two detection (MultiviewX and Wildtrack) datasets. The results seem to indicate that the paper is able to achieve similar classification and detection performance using a small number of views, when comapred with models using all views.

**Strengths:**

- The idea of the paper appears useful (cutting down on views)
- The results indicate that a decent performance can be attained while using a small number of camera views

**Weaknesses:**

- The method aspect of the paper is not fully clear. I request the authors to comment on following aspects:

(a) As far as I understood, MVSelect performance view selection for each frame for multiview detection problem? Is that right? if Yes, the selection process would also add latency? Can this be discussed in detail, because it is not clear in the current version?

(b) Does the camera selection process needs to compute features?

(c) Did you try a density based approach? View with more pedestrians, can be a better view?

(d) Please add few lines on the FOV maximization strategy in the paper itself. A brief description is warranted, as it crucial to understand if it was done correctly in your implementation.

(e) For multi-view classification, did you try methods like views which are 120 degree apart etc? Rather than mere random selection.


- Wildtrack and MultiviewX datasets are extremely small with significant train and test overlap (a huge concern given that the paper is reviewed in a ML forum). That is a major concern to draw any conclusions? Cross dataset experiments would be crucial to understand, if the method effectively works. The used classification datasets are also small scale. Hence, the paper lacks sufficient evidence.

- Some other minor concerns

(a) Figure4, mention it as modified implementation of MVDet. As mentioned in your paper, the original paper does concatenation. Thats a minor but a huge difference from the design perspective (view invariance).

(b) There are some typos e.g. Figure Figure5 caption: accuracy fir the task network

**Questions:**

Please address the points raised above, especially focusing on the lack of clarity in the method section.